# Enhancing Sentiment Analysis in Financial Markets Using RNNs and Word Embeddings

**Ethan Wei Yuxin, Chan Thong Fong, Liu Yiyang**
Tsinghua University
weiyx24@mails.tsinghua.edu.cn, ctf24@mails.tsinghua.edu.cn, lyy24@mails.tsinghua.edu.cn

## Abstract

This project centers on enhancing sentiment analysis in financial markets by classifying and interpreting sentiment from sources such as news articles and social media. Sentiment analysis offers unique insights valuable for financial predictions as it often aligns closely with stock movements [1, 2]. Using Recurrent Neural Networks (RNNs), particularly Long Short-Term Memory (LSTMs) and Gated Recurrent Units (GRUs), along with word embeddings, this project aims to capture sentiment trends over time, which may benefit investors by improving risk management and investment strategies.

## 1 Introduction

Sentiment analysis in financial markets is increasingly valuable as a tool for forecasting, particularly due to its alignment with stock market movements [1, 2]. Traditional forecasting methods, including ARIMA and Support Vector Machines, often lack the capability to capture nuanced emotional shifts that impact market volatility. Sentiment analysis, particularly with RNNs, offers unique insights by analyzing sources such as executives' social media posts and news headlines. Through RNN architectures, especially LSTM networks and GRUs, this project aims to classify and interpret sentiment in a way that benefits investors by improving risk management and investment strategies.

## 2 Problem Definition

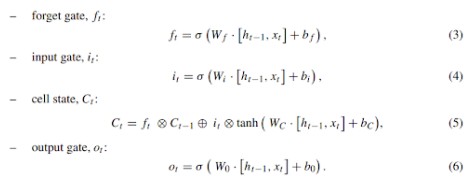

– forget gate, $f_t$:
$$f_t = \sigma\left(W_f \cdot [h_{t-1}, x_t] + b_f\right), \qquad (3)$$

– input gate, $i_t$:
$$i_t = \sigma\left(W_i \cdot [h_{t-1}, x_t] + b_i\right), \qquad (4)$$

– cell state, $C_t$:
$$C_t = f_t \otimes C_{t-1} \oplus i_t \otimes \tanh\left(W_C \cdot [h_{t-1}, x_t] + b_C\right), \qquad (5)$$

– output gate, $o_t$:
$$o_t = \sigma\left(W_0 \cdot [h_{t-1}, x_t] + b_0\right). \qquad (6)$$

Figure 1: LSTM Architecture Overview

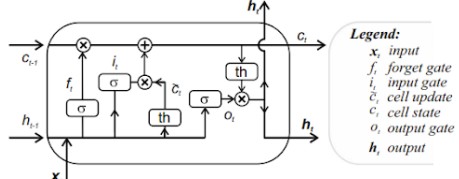

Figure 2: LSTM Unit Details

The problem this project addresses is time series forecasting using LSTM models, where the primary focus is to classify sentiment in financial text data and examine its correlation with market trends over time. Sentiment analysis plays a crucial role in understanding investor sentiment, which impacts stock performance. Using techniques like word embeddings, such as GloVe or word2vec, the project aims to capture these sentiment trends accurately and comprehensively [3].

Submitted to 38th Conference on Neural Information Processing Systems (NeurIPS 2024). Do not distribute.

# 3 Related Work

Jin, Yang, and Liu [4] applied a Convolutional Neural Network (CNN) with word2vec to map words into a multidimensional space and combined it with an LSTM-based RNN. Through Empirical Modal Decomposition (EMD) and an Attention Mechanism, their model improved prediction accuracy by 11.01% over a base LSTM model. Similarly, Shankar, Rohith, and Karthikeyan [10] developed an Ensemble Model that utilized LSTM, ARIMA, and Twitter sentiment analysis, achieving high accuracy in predicting market prices. They conducted sentiment analysis through binary classification and tested their model on data spanning from 2020 to 2024, demonstrating its efficacy in capturing public sentiment.

# 4 Proposed Method

The proposed method involves using RNNs, particularly LSTMs and GRUs, to classify and interpret sentiment from news articles and social media. This approach leverages word embeddings, such as GloVe and word2vec, to capture sequential dependencies and semantic meaning within text data. The primary goal is to analyze public and market sentiment, thus aiding in evaluating market mood and predicting reactions to news events.

If time permits, the project will extend to predict stock prices by applying RNNs to historical data sourced from financial APIs, such as Yahoo Finance or Alpha Vantage. This addition showcases the versatility of RNNs across both textual and numerical time-series data.

# 5 Methodology

The process we propose involves the following steps:

1. **Preprocessing the text data:** This involves cleaning, tokenizing, and normalizing data from sources such as news articles and social media.

2. **Loading pretrained word embeddings:** Using embeddings like GloVe or word2vec to represent text data in a multidimensional space, allowing the model to capture semantic meaning effectively.

3. **Defining the RNN architecture:** Building a model using LSTM or GRU units, with an architecture that suits sentiment analysis and stock price prediction tasks.

4. **Training the model:** Training the RNN model on sentiment-labeled data to allow it to learn sentiment trends.

5. **Evaluating the model's performance:** Assessing the model's accuracy in classifying sentiment and predicting stock prices.

6. **Iterating the process:** Repeating the steps above and refining the architecture until performance converges to a satisfactory level.

This methodology addresses common challenges in financial forecasting, such as overfitting, sensitivity to noise, and handling highly volatile markets. Our approach applies RNNs to both sentiment analysis and historical price data, creating a two-pronged strategy for generating a unified buy/sell signal.

# 6 Conclusion

In summary, this project aims to enhance sentiment analysis in financial markets by integrating advanced RNN architectures and word embeddings to interpret and analyze market sentiment. This approach not only facilitates an understanding of market mood but also provides a foundation for predicting stock market reactions to news events, potentially offering investors improved tools for managing risk and devising investment strategies.

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
