# OpenReview forum: "[Proposal-ML]Enhancing Sentiment Analysis in Financial Markets Using RNNs and Word Embeddings"
_tsinghua.edu.cn/THU/2024/Fall/AML — THU 2024 Fall AML Submission_

### Official Review · ~Guilherme_Félix_Diogo1 · 2024-11-06
**Complete proposal, evaluating the performance could be more complete**

**Rating:** 8
**Confidence:** 4

**Review:**

This proposal presents the use of sentiment analysis for financial markets using RNN architectures, specifically LSTM and GRU networks, combined with word embeddings. This project shows coherent reasoning in that it focus on stock market index movements and sentiment analysis in stock market and presents a framework that integrates text and numbers. The usage of RNNs together with embedding such as GloVe and word2vec is appropriate given the current state of affairs as the research tackle issues such as attentiveness to market fluctuations and noise. Nevertheless, while the overall methodology is comprehensive, the proposal could maybe include more information on how it will evaluate the performance of the two tasks: the sentiment classification and the stock prediction.

---

### Official Review · ~Huajun_Bai1 · 2024-11-07
**Relation of sentiment and stock prices?**

**Rating:** 6
**Confidence:** 3

**Review:**

3. Defining the RNN architecture: Building a model using LSTM or GRU units, with an
 architecture that suits sentiment analysis and stock price prediction tasks.
 4. Training the model: Training the RNN model on sentiment-labeled data to allow it to
 learn sentiment trends.

I think this part needs more explanation: sentiment analysis is a classification task and stock price prediction is a numerical prediction task. The architecture that fits both tasks is not intuitive to me.

---

### Official Review · ~Lily_Sheng1 · 2024-11-08
**Submission 28 Review**

**Rating:** 9
**Confidence:** 4

**Review:**

This work uses RNNs, particularly LSTMs and GRUs, to conduct sentiment analysis in financial markets. The use of word embeddings such as GloVe and word2vec enhances the model's ability to capture semantic meaning.

Pros:
1. The work has clearly listed methodology and approaches.
2. The applications of LSTM and GRU is well-suited to capturing dependencies in financial data.

Cons:
1. The accuracy of the sentiment analysis depends on the effectiveness of data processing, which may be challenging given the varied formats and the quality of text data from online sources.
2. There is no clearly defined baseline or metric for comparison of the approach.

---

### Official Review · ~Yunghwei_Lai1 · 2024-11-08

**Rating:** 7
**Confidence:** 3

**Review:**

While RNNs, LSTMs, and GRUs are good choices, they are relatively traditional models for sentiment analysis. Implementing more recent architectures like Transformer-based models (for example BERT, RoBERTa) could yield better accuracy and a more nuanced understanding of sentiment. The project could be strengthened by mentioning details like the training data's quality, model evaluation metrics, and potential benchmarking against existing sentiment analysis solutions for finance.

---

### Official Review · ~Jinsong_Xiao1 · 2024-11-09
**review for proposal 28**

**Rating:** 7
**Confidence:** 4

**Review:**

This proposal focuses on using RNN, specifically LSTM and GRU, along with word embeddings to enhance sentiment analysis in financial markets. By analyzing sentiment trends from news articles and social media, the authors aim to provide predictive insights to aid investors with risk management and investment strategies.

Strengths

- Clear Methodology: The authors present a straightforward approach and detailing steps.

- Comprehensive Related Work: The proposal draws from diverse sources, positioning the study within the context of existing literature.

Weaknesses

- Limited Innovation: The proposed approach does not offer notable methodological advancements beyond existing models.

- From the perspective of overall data, is the training data that only considers the emotions of the public on the Internet biased?

- Whether you can consider using the LLM method learned in the course to complete the task.

---

### Official Review · ~André_Moreira_Leal_Leonor1 · 2024-11-09
**Promising Approach to Financial Sentiment Analysis Using RNNs and Word Embeddings**

**Rating:** 9
**Confidence:** 4

**Review:**

This proposal provides a promising approach for the application of RNNs, specifically LSTMs and GRUs, combined with word embeddings like GloVe and word2vec, for sentiment analysis in financial markets. With a focus on sentiment classification from news and social media, the approach delineated herein is meant to grasp sentiment trends that move in concert with market dynamics in an effort to potentially help with risk management and investment strategy. The approach addresses problems of volatility and noise, forming a well-structured base for developing tools that would be immensely helpful in improving financial forecasting.

---

### Official Review · ~Renrui_Tian1 · 2024-11-10
**Relevant Topic but Limited Novelty and Clarity in Framework**

**Rating:** 6
**Confidence:** 3

**Review:**

**Strengths**:
* **Relevant Topic**: Sentiment analysis in financial markets is a timely and valuable area of research, with potential to improve forecasting and risk management.

**Weaknesses**:
* **Lack of Novelty and Contributions**: The proposal does not clearly articulate the novel contributions of the proposed method. It is crucial to differentiate your approach from prior research and highlight your unique contributions.
* **Limited Problem Definition**: The problem definition does not clarify the specific objectives and desired outcomes of the framework.

**Overall**: While the proposal delves into an important topic with potential applications in financial forecasting, further refinement is needed to clarify its unique contributions and specify the framework’s objectives.

---

### Official Review · ~Chenxi_Hu4 · 2024-11-11
**Lack Novelty of Methodology**

**Rating:** 7
**Confidence:** 4

**Review:**

The proposal presents a promising approach to enhancing sentiment analysis in financial markets using RNNs and word embeddings. The methodology is well-structured, and the proposed two-pronged strategy offers potential for generating valuable insights. However, the proposal lacks clarity on the novel contributions and specific details like data sources and evaluation metrics. Further refinement is needed to distinguish the approach from prior research.

---

### Official Review · ~Cheng_Gao2 · 2024-11-12
**Review for Enhancing Sentiment Analysis in Financial Markets Using RNNs and Word Embeddings**

**Rating:** 7
**Confidence:** 4

**Review:**

Strengths:

- The methodology and task definition are clearly presented.

Weaknesses:

- The authors propose using LSTM, RNN, and word embeddings to process financial markets data, but: (1) These techniques are a little bit outdated—methods like word2vec and GloVe are from around a decade ago and may not perform optimally. I recommend you to consider alternatives like BERT or using the OpenAI embedding model API instead; (2) The proposed approach lacks methods tailored specifically for financial market data, which may limit performance. Post-training the model for financial data might enhance results.
- While the methodology and task definition are relatively comprehensive, they would benefit from further depth. For instance, more detail on how LSTM, RNN, and word embedding techniques will be combined, as well as the intended output format, would strengthen this proposal.

---

### Official Review · ~Suraj_Joshi2 · 2024-11-12
**Review on Enhacing Sentiment Analysis for Financial Markets**

**Rating:** 9
**Confidence:** 4

**Review:**

This proposal proposes a method to extract market sentiment about a stock using various textual information sources about that stock. The market sentiment of a stock can be one of the major factor for determining the price of the stock. This proposal leverages RNN models which are well suited for sequence modeling tasks like language modeling.

However, I do feel some of the information is still missing in the proposal that would add a great clarity on the current method. The missing points in the proposal are listed as:
1. The sentiment of a stock is volatile and it hence it can only be useful to predict stock prices in the short term. In the longer run, various other factors of the company play important role in determining stock price., how can this issue be addressed, sometimes market sentiment change within a second and markets react immediately.
2.  Authors mention- "this project aims to classify and interpret sentiment in a way that benefits investors by improving risk management and investment strategies" but how do you encode this objective while training an RNN model?
3. Transformer based models have been demonstrating a great performance for text classification tasks, and they can also be parallelized for faster training, so why you guys don't consider Transformer based models for this project?

---

### Official Review · ~Isak_Tønnesen1 · 2024-11-12
**Review of "Enhancing Sentiment Analysis in Financial Markets Using RNNs and Word Embeddings"**

**Rating:** 7
**Confidence:** 4

**Review:**

The proposal presents a methodical approach to financial market sentiment analysis using RNNs and word embeddings. The methodology is well-structured and the combination of sentiment analysis with historical price data shows promise for practical applications.

Strengths:
- Clear, step-by-step methodology
- Practical application potential for investors
- Sound technical foundation with RNN/LSTM architecture
- Integration of both textual and numerical data

Weaknesses:
- Limited technical innovation beyond established methods
- Lacks specific evaluation metrics for measuring success
- Insufficient discussion of data quality and bias
- No consideration of modern transformer-based models

While the approach is solid, incorporating newer architectures like BERT or RoBERTa or API to existing models could significantly strengthen the proposal. Additionally, more attention to evaluation metrics and data quality would improve its robustness.

---

### Official Review · ~Han-Xi_Zhu1 · 2024-11-12
**Review on Enhacing Sentiment Analysis for Financial Markets**

**Rating:** 7
**Confidence:** 4

**Review:**

The proposal focuses on financial market sentiment analysis, which is an area that has a significant impact on financial forecasting and decision-making. With the increasing influence of social media and news articles on market sentiment, this work has practical application value.
## Strength
1. The proposal outlines a clear step-by-step methodology, including data preprocessing, model training, and evaluation, which aids in understanding the research process.
2. The paper is well-organised with it clear description on methodology and clear definition on the focused task.
## Weakness
1. The proposal lacks details on the data sources to be used and how the quality and relevance of the data will be ensured. The quality of the data directly impacts the accuracy and validity of the model.
2. While the proposal mentions evaluating model performance, it does not specify the metrics and benchmarks that will be used to assess the model's accuracy and efficiency.